# Rapid syntheses of *N*-fused heterocycles via acyl-transfer in heteroaryl ketones

Dan Ye[1,3], Hong Lu [1,3], Yi He[1], Zhaojing Zheng[2], Jinghao Wu[1] & Hao Wei [1✉]

The wide-ranging potencies of bioactive *N*-fused heterocycles inspire the development of synthetic transformations that simplify preparation of their complex, diverse structural motifs. Heteroaryl ketones are ubiquitous, readily available, and inexpensive molecular scaffolds, and are thus synthetically appealing as precursors in preparing *N*-fused hetero-cycles via intramolecular acyl-transfer. To best of our knowledge, acyl-transfer of unstrained heteroaryl ketones remains to be demonstrated. Here, we show an acyl transfer-annulation to convert heteroaryl ketones to *N*-fused heterocycles. Driven via aromatisation, the acyl of a heteroaryl ketone can be transferred from the carbon to the nitrogen of the corresponding heterocycle. The reaction commences with the spiroannulation of a heteroaryl ketone and an alkyl bromide, with the resulting spirocyclic intermediate undergoing aromatisation-driven intramolecular acyl transfer. The reaction conditions are optimised, with the reaction exhi-biting a broad substrate scope in terms of the ketone and alkyl bromide. The utility of this protocol is further demonstrated via application to complex natural products and drug derivatives to yield heavily functionalised *N*-fused heterocycles.

[1] Key Laboratory of Synthetic and Natural Functional Molecule of the Ministry of Education, College of Chemistry & Materials Science, Northwest University, 710069 Xi'an, China. [2] College of Food Science and Technology, Northwest University, 710069 Xi'an, China. [3] These authors contributed equally: Dan Ye, Hong Lu. ✉email: haow@nwu.edu.cn

*N*-fused heterocyclic compounds, such as pharmaceuticals, agrochemicals, plastics, and dyes (Fig. 1a), are integrated into everyday life[1–6]. Big data analysis shows that heterocycle synthesis is one of the most common reactions in the field of medicinal chemistry[7,8]. Among the best-selling therapeutics, almost a third contain fused heterocyclic structures[9]. Due to the high value of *N*-fused heterocycles, their novel, effective, flexible, general syntheses require investigation[10–12].

Acyl transfer is a critical process in various biological transformations[13]. In the field of organic synthesis, acyl transfer is frequently used in formation carbonyl compounds[14–18]. A typical acyl transfer employs a reactive carboxylic acid derivative (e.g. an acyl chloride or a thioester) as an acyl source. However, whether relatively inert ketones may serve as acyl transfer agents remains unclear?

Ketones are ubiquitous functional groups that not only occur widely in drug molecules and natural products but also act as bulk feedstocks in the syntheses of fine chemicals and materials. They are stable, non-toxic, and simple to prepare via various methods, rendering them ideal synthetic precursors[19]. If intramolecular acyl transfer of heteroaryl ketones can be realised, a transfer-annulation strategy may be employed in *N*-fused heterocycle preparation (Fig. 1b). However, owing to the kinetic inertness of C–C bonds, acyl transfer of ketones largely focuses on highly strained ketones[20–26]. For unstrained ketones[27–32], the most common strategy involves using directing groups to form of a stable chelate (Fig. 1c)[33–40]. Although effective, the use of directing groups complicates the overall synthesis and limits the scope of the accessible products. Hence, a acyl transfer of unstrained ketones for use in *N*-fused heterocycle synthesis is warranted.

Aromatisation, which enables delocalisation of electron density, stabilising the molecule[41], is a critical thermodynamic driving force in the field of organic chemistry[42–45], e.g. aromatisation-driven deacylations of ketones are prominent bond-cleavage strategies[46–48]. Therefore, we conceived a approach for the acyl transfer of unstrained heteroaryl ketones driven by aromatisation of a pre-aromatic intermediate (Fig. 1d). This strategy may be

suitable for use in the syntheses of *N*-fused heterocycles, and, critically, the directing group is no longer required. The next challenge in this strategy is the in situ formation of special, high-energy, pre-aromatic substrates. Transition metal-catalysed dearomatisation is a straightforward strategy to prepare spirocyclic scaffolds[49–52]. The spirocyclic intermediates, which are formed in situ from readily available heteroaryl ketones via dearomatisations[53–56], should serve as pre-aromatic precursors to facilitate rearrangement (Fig. 1d). This likely involves a Pd-catalysed dearomative spirocyclisation of a heteroaryl ketone with an alkyl bromide to generate a pre-aromatic intermediate (**A**), which is then intramolecularly trapped by the heterocyclic nitrogen[57–61]. The resulting intermediate (**B**) may subsequently lose a hydrogen, restoring aromaticity to yield the fused heterocyclic product.

Here, we report an acyl transfer-annulation of heteroaryl ketones driven by aromatisation. This method is operationally simple, scalable, and applicable to late-stage modifications of natural products and drug derivatives, which make it a valuable method for the synthesis of organic *N*-fused heterocycles.

## Results

**Reaction optimisation.** To explore this strategy, we initially used a heteroaryl ketone with a tethered olefin (**1**), which was prepared in one step using commercially available benzimidazole and 2-vinylbenzoyl chloride, as a model substrate. Because of the unique properties of difluoromethylene group (CF$_2$) and its critical applications in medicinal chemistry[62–64], ethyl bromodifluoroacetate (BrCF$_2$COOEt) was employed as the coupling partner. After systematic screening, the desired rearrangement product (**2**) is obtained in a 90% yield using PdCl$_2$ in combination with 1,1-bis(diphenylphosphino)pentane (dpppent, **L1**) as the ligand and Na$_2$CO$_3$ as the base in dioxane/tetrahydrofuran (THF) (Table 1, entry 1). The structure of **2** was unambiguously determined by X-ray crystallography. In addition, the Pd catalyst appears to be critical in this reaction. Using Pd(OAc)$_2$ or Pd$_2$(dba)$_3$ (dba = dibenzylideneacetone) as the catalyst results in much lower yields (Table 1, entries 2–3), and other metals, such as NiCl$_2$ and FeCl$_2$, are

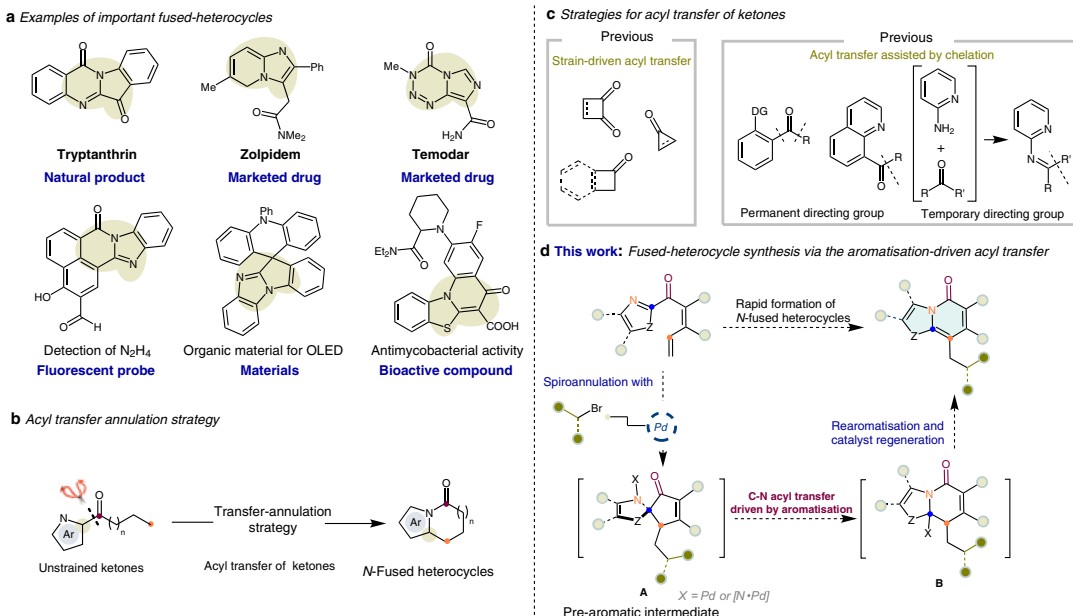

**Fig. 1 Examples of critial fused-heterocycles and our reaction design. a** *N*-fused heterocycles are ubiquitous within critical molecules, including biologically active natural and synthetic compounds and fine chemicals for use in functional materials. **b** Transfer-annulation strategy for synthesis of *N*-fused heterocycles. **c** Different strategies used in acyl transfer of ketones. **d** Fused heterocycle synthesis in this study via aromatisation-driven acyl tranfer of heteroaryl ketones with alkyl bromides.

**Table 1 Screening of reaction conditions.**

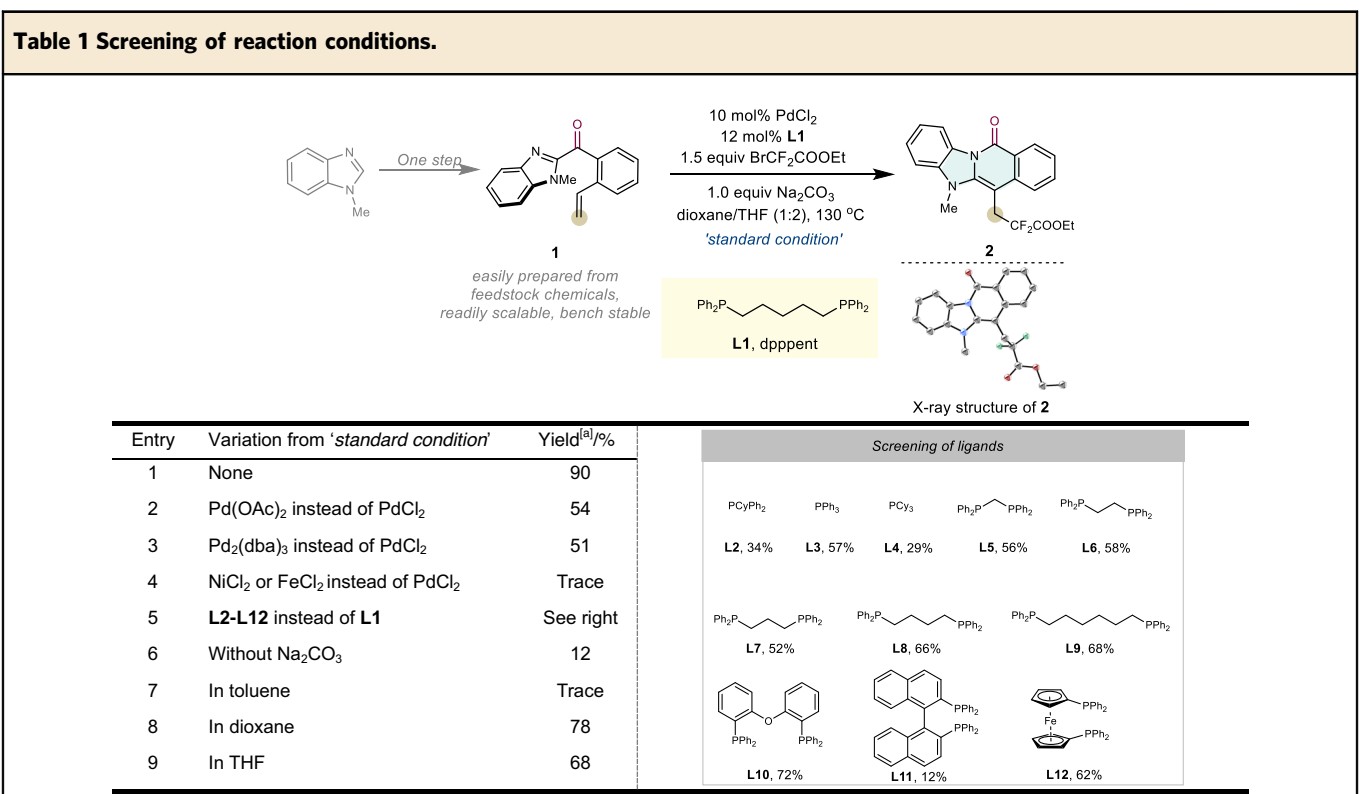

| Entry | Variation from 'standard condition' | Yield[a]/% |
|---|---|---|
| 1 | None | 90 |
| 2 | Pd(OAc)$_2$ instead of PdCl$_2$ | 54 |
| 3 | Pd$_2$(dba)$_3$ instead of PdCl$_2$ | 51 |
| 4 | NiCl$_2$ or FeCl$_2$ instead of PdCl$_2$ | Trace |
| 5 | L2-L12 instead of L1 | See right |
| 6 | Without Na$_2$CO$_3$ | 12 |
| 7 | In toluene | Trace |
| 8 | In dioxane | 78 |
| 9 | In THF | 68 |

Screening of ligands

L2, 34%  L3, 57%  L4, 29%  L5, 56%  L6, 58%

L7, 52%  L8, 66%  L9, 68%

L10, 72%  L11, 12%  L12, 62%

Unless otherwise specified, all reactions were carried out using **1** (0.1 mmol) and ethyl bromodifluoroacetate (0.15 mmol, 1.5 equiv), with 10 mol% PdCl$_2$, 12 mol% **L1** and Na$_2$CO$_3$ (1.0 equiv) in dioxane/THF (1:2) at 130 °C for 24 h. The CCDC number of **2** is 2116750.
[a]Isolated yields after chromatography.

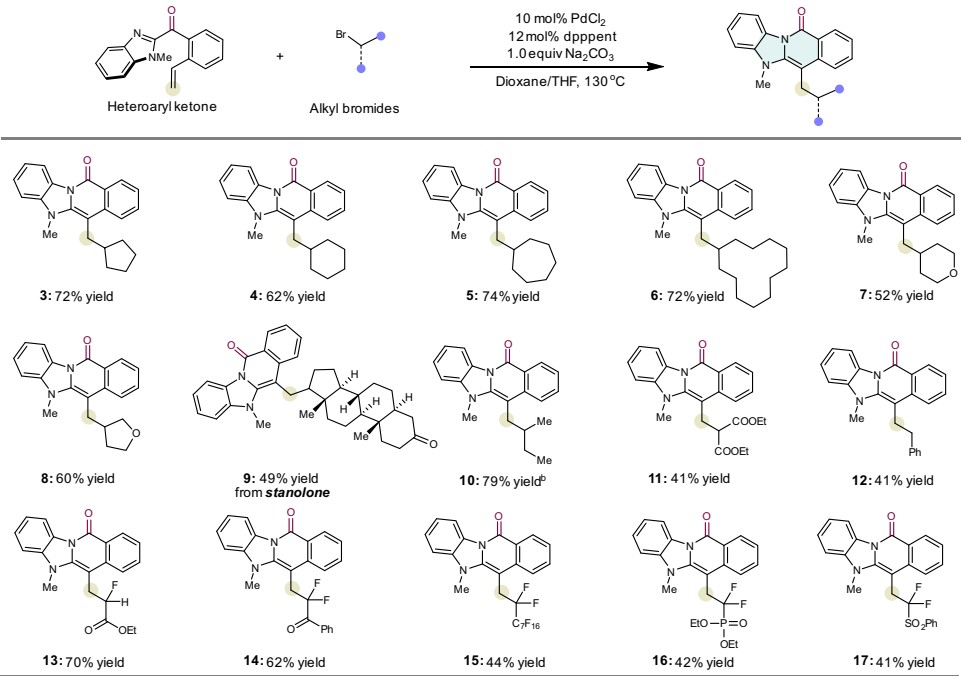

**Fig. 2 Substrate scope of alkyl bromides.** Unless otherwise specified, all the reactions were carried out using ketone **1** (0.1 mmol, 1.0 equiv) and alkyl bromide (0.15 mmol, 1.5 equiv.), PdCl$_2$ (10 mol%), dpppent (12 mol%) and Na$_2$CO$_3$ (1.0 equiv) in dioxane/THF (1:2) at 130 °C. Isolated yields after chromatography are shown.

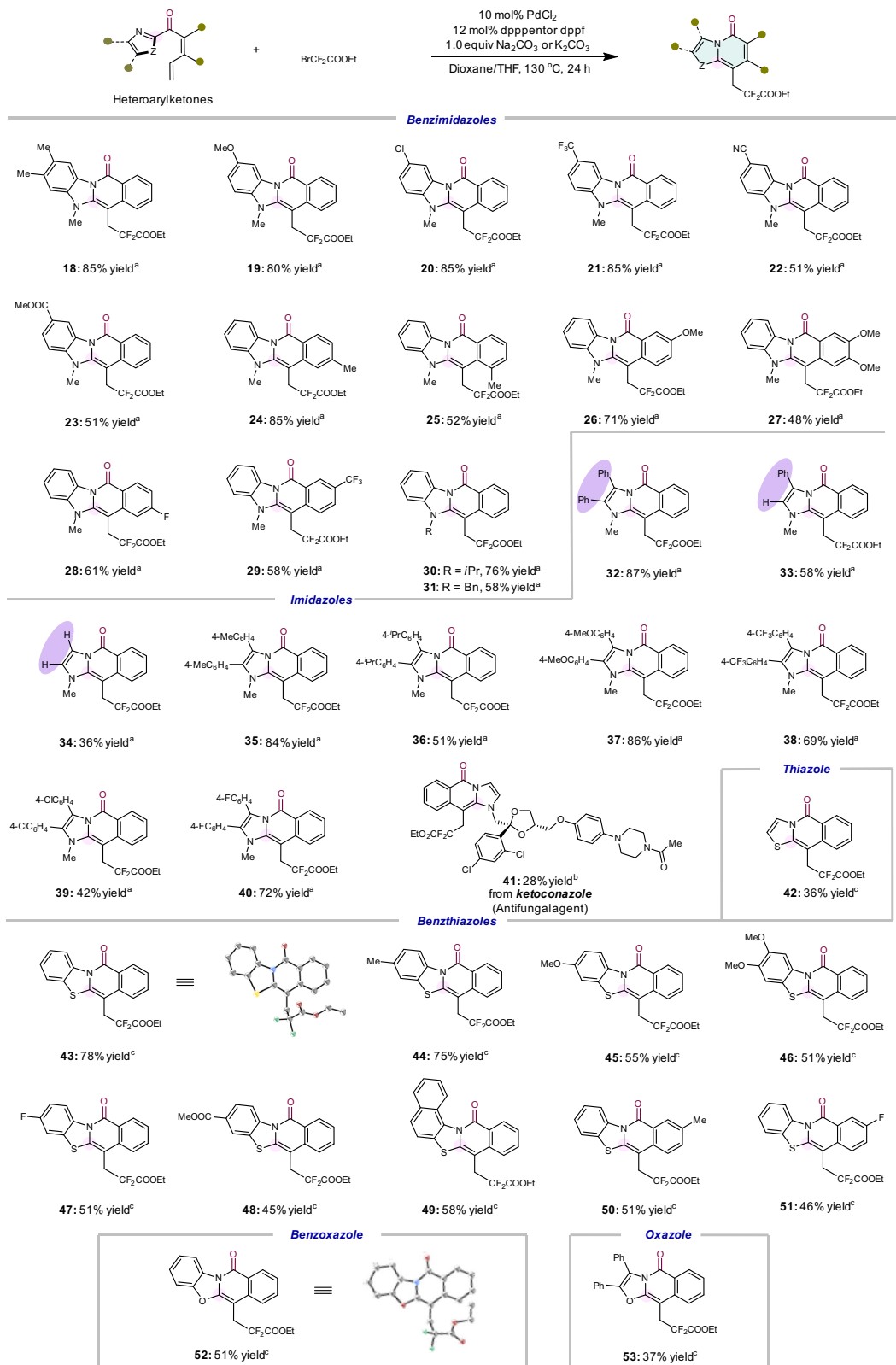

**Fig. 3 Substrate scope of heteroaryl ketones.** Isolated yields after chromatography are shown. The CCDC number of **43** is 2116753, **52** is 2116752. [a]The reaction was performed under optimised condition A: ketone **1** (0.1 mmol, 1.0 equiv) and ethyl bromodifluoroacetate (0.15 mmol, 1.5 equiv), PdCl₂ (10 mol %), dpppent (12 mol%) and Na₂CO₃ (1.0 equiv) in dioxane/THF (1:2) at 120 °C for 24 h. [b]The reaction was conducted under optimised condition A with a slight modification: bis(2-diphenylphosphinophenyl)ether (DPEPhos) (12 mol%) was used as ligand during the reaction. [c]The reaction was performed under optimised condition B: ketone **1** (0.1 mmol, 1.0 equiv) and ethyl bromodifluoroacetate (0.15 mmol, 1.5 equiv), PdCl₂ (10 mol%), dppf (12 mol%) and K₂CO₃ (1.0 equiv) in dioxane/THF (1:1) at 130 °C for 24 h. dppf = 1,1'-bis(diphenylphosphino)ferrocene.

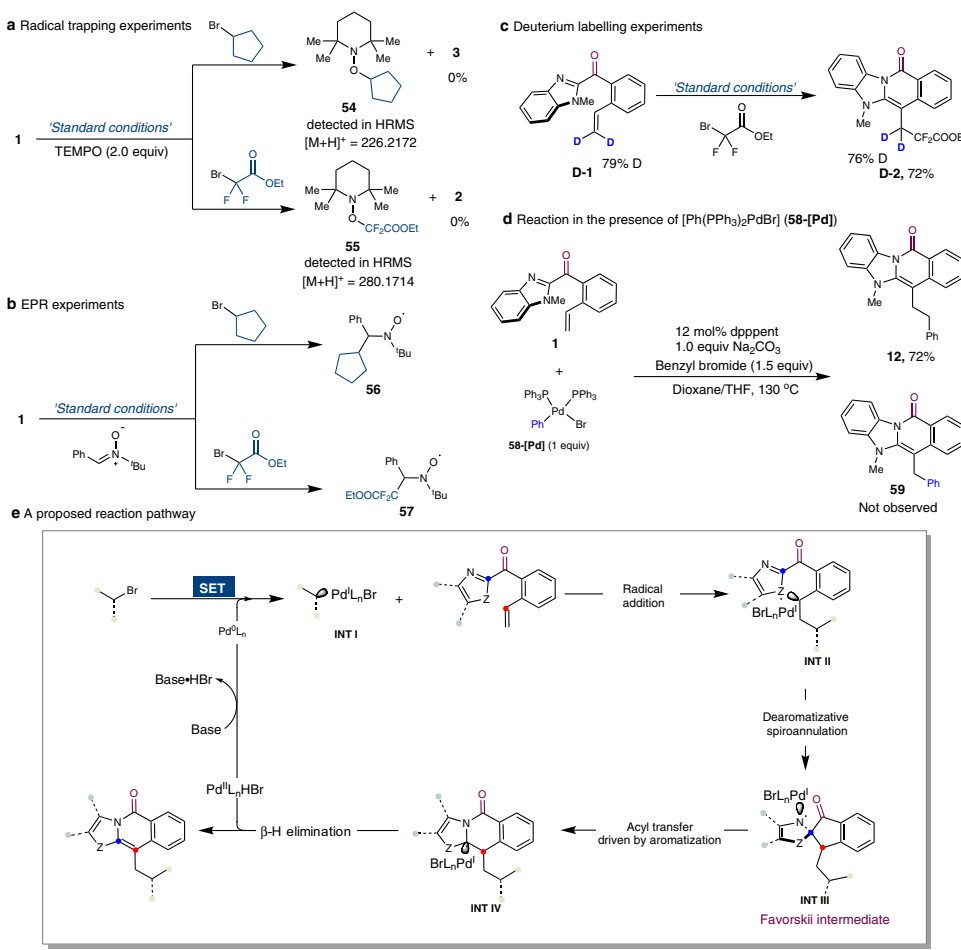

**Fig. 4 Mechanistic studies. a** Radical trapping study using TEMPO showing that alkyl radical species are involved in the reaction. **b** EPR studies also suggest that this reaction may involve alkyl radicals. **c** Deuterium labelling studies. **d** Reaction of **1** with benzyl bromide in the presence of [Ph(PPh₃)₂PdBr] (**58-[Pd]**). **e** A proposed reaction pathway.

completely ineffective (Table 1, entry 4). A study of the ligand effect further suggests that bidentate phosphine ligands are generally superior, with the yield increasing with the increasing bite angle of the phosphine employed, and **L1** is the only ligand that generates full conversion with the optimal yield (Table 1, entry 5). The addition of a base improves the reaction outcome appreciably, likely by neutralising the in situ-generated HBr (Table 1, entry 6). A survey of different solvents reveals that dioxane and THF are individually good, albeit generating slightly lower yields than that obtained using the mixture (Table 1, entries 7–9).

**Substrate scope**. With the conditions determined, the scope of alkyl bromides was examined first (Fig. 2). Ketone **1** is success-fully coupled with various alkyl bromides, with 5-, 6-, 7-, or 12-membered cycloalkyls (**3–6**) generating good yields of the desired coupling products. Heterocyclic bromides, with moieties such as tetrahydropyrane (**7**) and THF (**8**), react smoothly, resulting in good yields. Remarkably, the polycyclic bromide derived from the natural steroid stanolone is also amenable to coupling under the reaction conditions (**9**). Linear alkyl bromides are also suitable for reaction (**10–12**). We then investigated substrates with a CF₂ group. Bromofluoroacetate, bromodifluoromethyl ketone, per-fluoroalkyl bromide, bromodifluoromethyl phosphonate, and bromodifluoromethyl sulfone effectively undergo the desired annulation (**13–17**).

We further explored the rearrangements of various heteroaryl ketones with bromodifluoroacetate (Fig. 3). The rearrangement

took place smoothly by using 2-acylimidazoles and 2-acylbenzimidazoles as substrates (**18–41**). Both electron-rich and deficient substrates are competent during the cyclization process. A range of functional groups are compatible, including aryl fluorides (**28** and **40**) and chlorides (**20** and **39**), trifluoromethyl (**21** and **38**), esters (**23**) and cyano (**22**), are all tolerated. Changing the nitrogen protecting group from methyl to isopropyl (**30**) and benzyl (**31**) did not significantly affect the reactivity.

Compared to the substrate with 4,5-diphenylimidazole (**32**), the reactions of 4-phenylimidazole (**33**) and imidazole (**34**) yield lower conversions, indicating that aromatisation is essential to promote the reaction. Marketed drug-derived ketones, such as ketoconazole (**41**), also react smoothly despite the presence of several other functional groups. Significantly, numerous sub-strates are synthesised via direct acylation of commercially available imidazoles or benzimidazoles, with the resulting ketones directly undergoing rearrangement, which further highlights the efficiency of this process. Further, we examined other types of heterocycles, which should yield different heterocyclic cores via rearrangement. Heterocycles such as thiazole (**42**), benzothiazoles (**43–51**), benzoxazole (**52**), and oxazole (**53**) may also be incorporated, yielding pharmaceutically interesting fused-ring skeletons[65,66].

**Mechanistic considerations**. A study was performed to investi-gate the reaction pathway. To determine whether an alkyl radical

**a**

**60**, 59%
from *ketoprofen*

**61**, 64%
from *fernoxone*

**62**, 58%
from *estrone*

**63**, 60%
from *oxaprozin*

**64** 48%
from *gemfibrozil*

**65**, 56%
from *naproxen*

**66**, 79%
from *nortropine*

**67**, 65%, dr = 2:1
from *L-menthol*

**68**, 49%
from *indometacin*

**69**, 70%
from *cholesterol*

**70**, 46%
from *ibuprofen*

**71**, 45%
from *flurbiprofen*

**72**, 54%
from *D-mannofuranose*

**73**, 56%
from *adapalene*

**b**

**74**, 86%

**75**, 53%

**76**, 48%

**77**, 68%

**Fig. 5 Synthetic applications. a** Using the tranfer-annulation strategy in the late-stage modifications of complex frameworks based on natural products and drug molecules. **b** Gram-scale synthesis and various useful transformations of **2**. The CCDC number of **74** is 2131840.

exists during this Pd-catalysed process, a radical inhibition study was performed. When 2,2,6,6-tetramethylpiperidinooxy (TEMPO) is added to the reaction mixture, it traps alkyl radicals, indicating that the reaction involves radical species (Fig. 4a). An electron paramagnetic resonance (EPR) study of the reaction of bromocyclopentane with the spin-trapping agent phenyl-N-tert-butylnitrone reveals the presence of spin adducts of the trapped alkyl radicals **56** and **57** (Fig. 4b), as indicated by the EPR spectrum (see supporting information). Deuterium labelling

studies were conducted using the heteroaryl ketone **D-1** (79% deuterium content) as a substrate under the optimised conditions, with a significant level of the deuterated product **D-2** (76% deuterium content) detected, suggesting that there were no reversible hydro-metallation in this process (Fig. 4c)[67,68]. Finally, we synthesised an aryl Pd complex (**58-[Pd]**), with **12** produced instead of **59** in the presence of **58-[Pd]**, benzyl bromide, and **1** (Fig. 4d). Therefore, the alkyl group of the fused heterocyclic product is not derived from the migratory insertion of the Pd(II)

complex. The proposed reaction pathway is thus shown in Fig. 4e. The reaction may be initiated by a single electron transfer between Pd(0) and the alkyl bromide, producing hybrid alkyl Pd(I)-radical species **INT I**. Subsequently, radical addition to the alkene occurs, leading to the hybrid benzylic radical **INT II**, which then undergoes dearomatisative-spirocyclisation to form the spiro-N-radical **INT III**. Aromatisation-driven intramolecular acyl transfer may then occur to form the alkyl radical **INT IV**. Subsequent β-H elimination at the latter yields the product with concomitant regeneration of the Pd catalyst. This proposed mechanism is also supported by X-ray photoelectron spectroscopy, which revealed the presence of three distinct Pd oxidation states (Pd(0), Pd(I), and Pd(II)) during the process, suggesting that Pd(I) species may be involved.

**Synthetic utility**. Further studies were conducted to demonstrate the viability of this acyl transfer-annulation strategy. The protocol was applied in the late-stage modifications of natural products and drug derivatives (Fig. 5a). Various complex molecules with diverse structural features, such as steroids (**62** and **69**), N-heteroarenes (oxazole **63** and indole **68**), alkaloids (**66**), and carbohydrates (**72**), are readily converted into the corresponding products in useful yields. This strategy provides a straightforward, versatile method of generating valuable N-fused heterocyclic moieties within complex molecules. Given the ubiquity of N-fused heterocycles in pharmaceuticals, this approach may be used in the field of medicinal chemistry.

To showcase the scalability of this process, a gram-scale reaction was carried out. Gratifyingly, a satisfactory 67% isolated yield (80% yield based on recovered **1**) of product **2** could be obtained without modification of the optimised conditions (Fig. 5b). The N-fused heterocyclic scaffold can readily undergo various transformations to access a range of synthetically useful scaffolds. For example, the bromination of **2** proceeded to afford **74**, excellent selectivity for the 9-position was observed, which allows follow-up fused heterocycle manipulations through cross-couplings. Treatment with mCPBA, deconstruction of N-fused heterocycle was observed, which afforded **75** in 53% yield. Diazidation product **76** was afforded in 48% yield via vicinal diazidation of olefin. Moreover, the ester moiety was smoothly reduced with NaBH$_4$, affording the corresponding alcohol **77** in 68% yield.

In conclusion, a synthetically useful, mechanistically intriguing intramolecular acyl transfer of heteroaryl ketones was developed, which was suitable for use in fused-ring synthesis. The formation of a high-energy pre-aromatic spirocyclic intermediate was critical in the successful transformation, with aromatisation the driving force that facilitated C–C bond cleavage. Given the ready availability of the ketone moiety, this strategy could be used to simplify the syntheses of complex N-fused heterocyclic systems, which are privileged structures within numerous biologically active compounds. Moreover, the protocol enabled the late-stage modifications of intricate natural products and drug derivatives and may thus facilitate heterocyclic drug discovery.

## Methods
**General condition A for transfer-annulation of heteroaryl ketones derived from (benzo)imidazoles**. In a nitrogen-filled glovebox, an oven-dried 10 mL sealed tube equipped with a Teflon-coated magnetic stir bar was charged successively with heteroaryl ketone **1** (0.1 mmol), alkyl bromide (0.15 mmol, 1.5 equiv), PdCl$_2$ (0.01 mmol, 10 mol%), dpppent (0.012 mmol, 12 mol%), Na$_2$CO$_3$ (0.1 mmol, 1.0 equiv) and dioxane/THF (1.0 mL, 1:2). The tube then was sealed with a Teflon screw cap, moved out of the glovebox, and placed on a hotplate pre-heated to 130 °C for 24–36 h. After completion of the reaction, the mixture was filtered through a thin pad of silica gel. The filter cake was washed with ethyl acetate and the combined filtrate was concentrated under vacuum. The residue was purified via silica gel chromatography.

**General condition B for transfer-annulation of heteroaryl ketones derived from (benzo)thiazoles and (benzo)oxazoles**. In a nitrogen-filled glovebox, an oven-dried 10 mL sealed tube equipped with a Teflon-coated magnetic stir bar was charged successively with heteroaryl ketone **1** (0.1 mmol), difluorobromoethyl ester (0.15 mmol, 1.5 equiv), PdCl$_2$ (0.01 mmol, 10 mol%), dppf (0.012 mmol, 12 mol%), K$_2$CO$_3$ (0.1 mmol, 1.0 equiv) and dioxane/THF (1.0 mL, 1:1). The tube then was sealed with a Teflon screw cap, moved out of the glovebox, and placed on a hotplate pre-heated to 120 °C for 24 h. After completion of the reaction, the mixture was filtered through a thin pad of silica gel. The filter cake was washed with ethyl acetate and the combined filtrate was concentrated under vacuum. The residue was purified via silica gel chromatography.

## Data availability

Data relating to the optimisation studies, mechanistic studies, general methods, and the characterisation data of materials and products, are available in the Supplementary Information. Crystallographic parameters for compounds **2**, **43**, **52** and **74** are available free of charge from the Cambridge Crystallographic Data Centre under CCDC 2116750 (**2**), 2116753 (**43**), 2116752 (**52**) and 2131840 (**74**). These data can be obtained free of charge from The Cambridge Crystallographic Data Centre via www.ccdc.cam.ac.uk/getstructures.

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

## Acknowledgements

We are grateful for the financial support from the National Natural Science Foundation of China (21971205), Key Research and Invention Program in Shaanxi Province of China (2021SF-299), Natural Science Basic Research Program of Shaanxi (2020JQ-574), Scientific Research Program of Shaanxi Education Department (No. 20JK0937) and Northwest University.

## Author contributions

H.W. conceived and designed the project and composed the paper. D.Y., H.L., Y.H. and J.W. conducted the experiments and analysed the data. H.L. and Z.Z. discussed the experimental results and commented on the paper. H.W. conducted general guidance, project directing, and paper revisions.

## Competing interests

The authors declare no competing interests.
