## [Peer Review File · Nature Communications]

REVIEWER COMMENTS

Reviewer #1 (Remarks to the Author):

In this manuscript, Wei et al. report a new method for the synthesis of N-fused heterocycles via Favorskii-type rearrangements. Unlike the classical Favorskii rearrangement, this strategy crucially eliminates the need for the formation of a highly strained cycloacetone intermediate. A series of transformations of readily available heteroaryl ketones are also competent for this reaction. The proposed reaction mechanism seems plausible, and in order to test the hypothesis, the authors endeavored to perform radical suppression studies to gain further understanding of the reaction mechanism. The reported transformation can be performed on a large scale, which is useful and important for the synthesis of a series of N-fused heterocycles. Overall, the results described here are interesting and worthy of publication, and with minor revisions the manuscript should be suitable for publication in Nature Communications.

Reviewer #2 (Remarks to the Author):

The manuscript by Wei et al describes an interesting intramolecular acyl transfer reaction through a radical mechanism. The work is important and useful for the synthesis of important heteroaryl frameworks, can be utilized in medicinal chemistry. The scope is good and control experiments are fine.

However, I have a main issue with the claims of this work and its title.

I read this manuscript with care and could not find why this is Favorskii like rearrangement. It is a simple acyl transfer reaction using a radical pathway. Nothing more than this, in my understanding. When someone states about a Favorskii rearrangement or semi-Favorskii rearrangement (e.g. <https://doi.org/10.1021/acs.orglett.8b00218>), the key intermediate cyclopropanone must be generated. Here in this reported reaction the authors are only correlating to the later part of the original Favorskii rearrangement. But this is not an appropriate claim. When we do not generate a cyclopropanone, and do an acyl transfer it cannot be called a Favorskii like rearrangement. The authors cite reference 32 and 33. It is clearly described in those papers about the cyclopropanone. Hence the title, introduction and entire claims of this reported work must be modified by removing any correlations with Favorskii rearrangement.

Moreover, this type of acyl transfer reactions is not new in the literature.

Hence my decision is rejection. And I suggest the authors to go for any synthetic organic chemistry specialized journal.

Reviewer #3 (Remarks to the Author):

Wei and co-workers reported a novel Favorskii-type rearrangement for the use in the skeletal remodeling of heteroaryl ketones, delivering a series of complex N-fused heterocyclic compounds. The formation of a high-energy pre-aromatic spirocyclic intermediate and aromatization as the driving force for C-C bond cleavage are the key points for the success of the reaction. This method could be used in the late-stage modifications of some natural products and drug derivatives. In addition, based on the control experiments, a possible reaction pathway was provided for the reaction. This report provides an efficient method for the preparation of N-fused heterocyclic compounds. On the basis of the above merits, I recommend this work to be published in the journal of "Nature Communications" after the following questions are addressed. 1) For the mechanistic studies, in Fig. 2d, catalytic amount of 58-[Pd] was used in the reaction, it is too less to detect the corresponding product 59. Stoichiometric 58-[Pd] should be used for the control experiment. 2) Grammatical and technical errors should be corrected. For example, line 58, "metnyl" should be "methyl"; line 148, "synthesised" should be "synthesized".

Responses to the reviewer's comments

Review 1.

In this manuscript, Wei et al. report a new method for the synthesis of N-fused heterocycles via Favorskii-type rearrangements. Unlike the classical Favorskii rearrangement, this strategy crucially eliminates the need for the formation of a highly strained cycloacetone intermediate. A series of transformations of readily available heteroaryl ketones are also competent for this reaction. The proposed reaction mechanism seems plausible, and in order to test the hypothesis, the authors endeavored to perform radical suppression studies to gain further understanding of the reaction mechanism. The reported transformation can be performed on a large scale, which is useful and important for the synthesis of a series of N-fused heterocycles. Overall, the results described here are interesting and worthy of publication, and with minor revisions the manuscript should be suitable for publication in Nature Communications.

Reviewer 2:

The manuscript by Wei et al describes an interesting intramolecular acyl transfer reaction through a radical mechanism. The work is important and useful for the synthesis of important heteroaryl frameworks, can be utilized in medicinal chemistry. The scope is good and control experiments are fine.

However, I have a main issue with the claims of this work and its title.

I read this manuscript with care and could not find why this is Favorskii like rearrangement. It is a simple acyl transfer reaction using a radical pathway. Nothing more than this, in my understanding. When someone states about a Favorskii rearrangement or semi-Favorskii rearrangement (e.g. <https://doi.org/10.1021/acs.orglett.8b00218>), the key intermediate cyclopropanone must be generated. Here in this reported reaction the authors are only correlating to the later part of the original Favorskii rearrangement. But this is not an appropriate claim. When we do not generate a cyclopropanone, and do an acyl transfer it cannot be called a Favorskii like rearrangement. The authors cite reference 32 and 33. It is clearly described in those papers about the cyclopropanone. Hence the title, introduction and entire claims of this reported work must be modified by removing any correlations with Favorskii rearrangement.

Moreover, this type of acyl transfer reactions is not new in the literature.

Hence my decision is rejection. And I suggest the authors to go for any synthetic organic chemistry specialized journal.

Answer: Thank you for the comment. The title, abstract, and introduction were rewritten, with the references to the Favorskii reaction and skeletal remodelling removed. We described the reaction in terms of acyl transfer.

Reviewer 3:

Wei and co-workers reported a novel Favorskii-type rearrangement for the use in the skeletal remodeling of heteroaryl ketones, delivering a series of complex N-fused heterocyclic compounds. The formation of a high-energy pre-aromatic spirocyclic intermediate and aromatization as the

driving force for C-C bond cleavage are the key points for the success of the reaction. This method could be used in the late-stage modifications of some natural products and drug derivatives. In addition, based on the control experiments, a possible reaction pathway was provided for the reaction. This report provides an efficient method for the preparation of N-fused heterocyclic compounds. On the basis of the above merits, I recommend this work to be published in the journal of "Nature Communications" after the following questions are addressed.

1) For the mechanistic studies, in Fig. 2d, catalytic amount of 58-[Pd] was used in the reaction, it is too less to detect the corresponding product 59. Stoichiometric 58-[Pd] should be used for the control experiment.

Answer: Thank you for the suggestion. We thus used a stoichiometric amount of 58-[Pd] in the control experiment, and Compound 59 was not observed during the reaction. Related studies were added to the revised manuscript.

2) Grammatical and technical errors should be corrected. For example, line 58, "metnyl" should be "methyl"; line 148, "synthesised" should be "synthesized".

Answer: Thank you for the suggestion. We revised the corresponding text

REVIEWERS' COMMENTS

Reviewer #3 (Remarks to the Author):

In the revised manuscript, Wei and co-workers have been addressed the reviewers' concerns. Now, I still recommend this work to be published in the journal of "Nature Communications".